# Do Rescuers’ Physiological Responses and Anxiety Influence Quality Resuscitation under Extreme Temperatures?

**DOI:** 10.3390/ijerph17124241

**Published:** 2020-06-14

**Authors:** José Luis Martin-Conty, Francisco Martin-Rodríguez, Juan José Criado-Álvarez, Carmen Romo Barrientos, Clara Maestre-Miquel, Antonio Viñuela, Begoña Polonio-López, Carlos Durantez-Fernández, Félix Marcos-Tejedor, Alicia Mohedano-Moriano

**Affiliations:** 1Faculty of Health Sciences, Universidad de Castilla la Mancha, 45600 Talavera de la Reina, Spain; JoseLuis.MartinConty@uclm.es (J.L.M.-C.); juanjose.criado@uclm.es (J.J.C.-Á.); Clara.maestre@uclm.es (C.M.-M.); Antonio.vinuela@uclm.es (A.V.); Begona.polonio@uclm.es (B.P.-L.); Carlos.durantez@uclm.es (C.D.-F.); Felix.MarcosTejedor@uclm.es (F.M.-T.); Alicia.Mohedano@uclm.es (A.M.-M.); 2Advanced Clinical Simulation Center, School of Medicine, Universidad de Valladolid, Avda. Ramón y Cajal, 7, 47005 Valladolid, Spain; 3Integrated Care Management of Talavera de la Reina, Health Services of Castilla-La Mancha (SESCAM), 45600 Talavera de la Reina, Toledo, Spain; mcromo@sescam.jccm.es

**Keywords:** cardiopulmonary resuscitation, simulation, emergency, anxiety, extreme temperatures

## Abstract

Teaching and training cardiopulmonary resuscitation (CPR) through simulation is a priority in Health Sciences degrees. Although CPR is taught as a simulation, it can still be stressful for the trainees since it resembles a real-life circumstance. The aim of this study was to assess the physiological effects and anxiety levels of health sciences undergraduates when faced with CPR process in different temperatures (room temperature, extremely cold, or extremely warm). This was a descriptive cross-sectional before–after study conducted during the 2018/2019 academic year with 59 students registered in the Faculty of Health Sciences of the Castilla-La Mancha University (UCLM). State Trait Anxiety Inventory (STAI) questionnaires were distributed among the students before and after the CPR simulation. We found greater level of situational anxiety in undergraduates faced with extreme adverse temperature scenarios (extreme heat and cold), especially in conditions of extreme heat compared to controlled environment (at room temperature). We discovered differences regarding sex, in which men scored 6.4 ± 5.55 points (STAI after CPR score) and women scored 10.4 ± 7.89 points (STAI after CPR score). Furthermore, there was less lactate in blood, before and during the event in individuals with anxiety. In addition, beginning in Minute 7, we observed a remarkable decrease (but not significant) in the performance of rescuers with anxiety. Programs targeted at promoting coping mechanisms to reduce anxiety before a critical clinic situation should be implemented in academic training.

## 1. Introduction

Learning through clinical simulation has important benefits for clinical practice in students of health sciences [1]. For this reason, cardiopulmonary resuscitation (CPR) is usually taught in medicine and nursing curricula through simulation. Clinical simulation laboratories provide real scenarios in contrast to conventional lecture rooms or other laboratories, because the available material is more advanced and the feedback with students is immediate [2]. What is more, a CPR simulation can be a stressful experience because it resembles a vital situation of extreme urgency, causing great anxiety for undergraduates, further than a basic training of theoretical concepts going with practical maneuvers (simulation learning) [3,4,5].

The main causes of stress, in these cases, are the pressure of the intervention, the critical condition of the patient, and aspects related to death and the process of dying [3,6,7]. Specifically, for students, other stressors related to the evaluation process have also been described, such as possible errors, presence of the teacher, inexperience, or a feeling of incompetence [2,8].

If students had previously participated in simulated experiences, their anxiety was lower in clinical practice [9], and their self-confidence and their ability to prioritize were higher [10,11].

Emergency teams commonly experience anxiety, described as an unpleasant sensation of intense and indeterminate fear that may affect appropriate decision making [5,12,13]. Anxiety can be defined as an adaptive emotional reaction that arises from doubtful or alarming situations, and that prepares the individual to respond and act appropriately. This state anxiety, characterized by subjective, consciously perceived feelings of attention and apprehension and by hyperactivity of the autonomic nervous system, appears when the individual overestimates the probability of a feared event, or simultaneously underestimates coping resources [14,15]. Reactions and feelings, as well as anxiety levels, during clinical simulation have been described previously in students of health sciences. These studies have focused on interpersonal and social stressors, such as the presence or absence of family members and observers [8,16], but always under normal temperature conditions. Thus far, no studies have been conducted in adverse temperature conditions of cold and heat. It is unknown how adverse temperature could affect the CPR process, and the derived anxiety on rescuers. This could be relevant for people who perform CPR in geographic places with adverse climate. We based our study on the possible differences between physiological effects and anxiety levels of rescuers performing CPR in extremely temperature vs. room temperature.

The aim of this study was to evaluate the physiological effects and anxiety levels of health sciences undergraduates when faced with performing CPR in different temperatures (room temperature, extremely cold or extremely warm temperatures).

## 2. Materials and Methods

### 2.1. Study Design

This was a descriptive cross-sectional before–after study [17] conducted from 6 to 10 May 2019 with students registered in nursing and occupational therapy degrees in the Faculty of Health Sciences of the Castilla-La Mancha University (UCLM) in the city of Talavera de la Reina (Toledo, Spain). Prior to conducting the CPR simulation, all participants had received a CPR seminar with dummies, taught by several teachers of nursing. All selected participants were aged between 18 and 30. They all had basic knowledge of CPR process, according to the American Heart Association or European Resuscitation Council training. The students were informed about the general objectives of the study and gave their informed consent. The study was approved by the Clinical Research Ethics Committee of Talavera de la Reina (Toledo) with number 178013/113. Details of the study design, statistical analysis plan, and baseline data are available online (doi.org/10.1186/ISRCTN10222040).

### 2.2. Population

To carry out the study, anyone aged 18–65 years with basic cardiopulmonary resuscitation knowledge (passed an accredited course or the equivalent training) was considered eligible. From a pool of all participants who showed interest in participating in the study, a random selection was performed. After randomization, the participants, if they did not present any exclusion criteria, signed the informed consent and carried out the study. The exclusion criteria were resting heart rate of >120 beats/minute (bpm) or <35 bpm; systolic or diastolic blood pressure >160 or <95 mmHg or systolic blood pressure <80 mmHg, respectively; blood sugar levels <65 mg/dL; severe visual or hearing impairment or any type of functional disorder hindering cardiopulmonary resuscitation maneuvers; major surgery in the last 30 days; epilepsy; diagnosed infections treated while the study was done; electrocardiogram with alterations (arrhythmia or changes in S-T segment); oxygen saturation <92%; body mass index >40 kg/m^2^; participant temperature >38 °C; acute-phase skin diseases; and systemic immunological diseases.

The sample size was 18 subjects per group (54 subjects); it was calculated as accepting a 0.05 alpha risk and a 0.2 beta risk with bilateral contrast. This indicated that it would be needed to detect a minimum difference of 0.1 between two groups, assuming there were three groups, with a standard deviation of 0.1.

### 2.3. Study Protocol

CPR was performed in three simulation scenarios with extreme weather: control (room temperature), extremely cold (at −35 °C), and hot (above 40 °C). Each space was adapted with the necessary system to simulate the different environments and three homogeneous groups were assigned with double blind masking: no participants knew in what kind of environment they were going to perform the test. The variable of temperature was encoded to guarantee the blind masking process. Each subject had the same probability of being included in any group, as a randomization sequence was generated using random numbers according to the gender stratification created by the computer with XLSTAT ^®^ BioMED software for Microsoft Excel® version 14.4.0. (Microsoft Inc., Redmond, USA).

Thirty minutes before taking the CPR test, all students completed the State Trait Anxiety Inventory (STAI) (State Anxiety (SA) and Trait Anxiety (TA)) questionnaires. They continued to the different CPR scenarios where they performed a test for 10 min, with a high-quality CPR simulator: Real CPR Help software installed in an R Series monitor-defibrillator (ZOLL Medical Corporation, Chelmsford, MA, USA) and CPR-D-padz^®^ defibrillation electrodes (ZOLL Medical Corporation, Chelmsford, MA, USA) (Real CPR Help^®^ technology gives you real-time feedback about the depth and frequency of CPR while applying it, which provides guidance on improving the quality of the CPR). 

In addition, the students’ respiratory rate (by observing thoracic expansion for 1 min) and heart rate (using pulse oximeter) before and after the CPR maneuvers were registered. Other parameters observed were temperature (tympanic thermometer Braun ThermoScan PRO 6000, Welch Allyn, Inc., Skaneateles Falls, USA), pH (using reactive saliva tests with color scale from brand Invigorated water pH), lactate obtained by a lactometer Accutrend Plus (Roche Diagnostics, Mannheim, Germany), with a measure range 0.8−21.7 mmol/L. The protocol to determine lactate from capillary consists of the following phases: Firstly, we switched on the lactometer and verified that reactive strips code coincides with code that appears on the screen’s device and the expiration date of the strips. Secondly, we extracted blood with a security lancet Solofix^®^ Safety (B. Braun, Melsungen, Germany) on index finger from the right hand. Finally, we put 15–40 μL of blood from capillary on the reactive strip, and after 60 s, we obtained the result. 

Once the test was completed, the participants filled in the STAI (TA and SA) questionnaires again. 

To assess the state of anxiety, the STAI questionnaire was used. All participants took the questionnaire to measure the anxiety before and after the CPR simulation. To prevent interference, the subjects were not informed that they were to complete the questionnaire again after finishing the simulation. The STAI is a questionnaire validated in Spanish and self-administered, conceived as a research instrument in the study of anxiety in healthy adults. It consists of two scales SA and TA of 20 questions each, which measure different, but related, aspects of anxiety. TA reflects a relatively stable, anxious propensity that characterizes individuals with a tendency to perceive situations as threatening. SA reflects the subjective and transient feelings of tension, apprehension, and fear that may vary over time and fluctuate in intensity. SA increases in response to different types of tension and decreases after relaxation techniques. The questionnaire provides a numerical value for TA and another for SA. The Total STAI is the sum of SA and TA [18,19,20]. The STAI is validated for the Spanish population and has a Cronbach’s alpha of 0.93 for TA and 0.92 for SA [21].

### 2.4. Data Analysis

Descriptive statistics (means ± standard deviation (SD)) were used for the quantitative and descriptive frequency variables. The categorical variables were described using absolute frequencies with their 95% confidence interval (CI 95%).

In the descriptive and inferential statistical analysis, the parameters were used according to the scale of the variable. The quantitative variables herein contemplated were: Trait STAI (basal anxiety), Trait STAI before, State STAI after, Total STAI before, and Total STAI after. The other quantitative variables related with the subjects’ physiological parameters were selected: lactate, heart rate, respiratory rate, temperature, and pH.

Physiological conditions at rest were evaluated in the different variables to learn subjects’ resting levels and to rule out any initial differences among the experimental groups. The resting condition was taken as the first minute recorded in the experimental protocol.

Data were checked to meet the normality condition by the Kolmogorov–Smirnov test. All the data presented a normal distribution; hence, non-parametric tests were not necessary. In the study, the female/male ratio was assessed by the χ^2^ test. A t-test was used to study the differences in paired variables. We studied by t-test: STAI variables according to sex and the physiological values related to anxiety level of participants. ANOVA statistical test was used to perform the data analysis of STAI variables according to experimental group. A 95% confidence level was established. The SPSS statistical package, v. 24 (SPSS Inc., Chicago, IL, USA) was employed.

## 3. Results

We included 59 subjects, of whom 27 (45.8%) were male and 32 (54.2%) female, with a mean age of 21.3 ± 2.7 years. According to academic degree, 83.1% were registered for the degree in nursing, and the rest (16.9%) for occupational therapy, with a predominance of first-year students (*n*=25; 42.4%). Overall, 72.9% of participants had previous knowledge about CPR, and the remaining 27.1% had to take a specific training at university to keep the requirements or inclusion criteria. There were no statistical differences in the quality of CPR performed by both groups. The experimental group was split into 19 students (32.2%) in the heat group and 20 (33.9%) in the cold group, while 20 subjects (33.9%) were assigned to the control group (Figure 1).

The distribution of the main variables according to sex is shown in Table 1. The only statistically significant difference (*p* < 0.05) appeared for the subsequent Total STAI according to sex, in which men scored 6.4 ± 5.55 points and women scored 10.4 ± 7.89 points. Twenty-five students (42.4%) showed a global anxiety in the initial phase (Total STAI before CPR) that persisted in the final or subsequent phase (Total STAI after CPR) (*p* < 0.05). There were no statistical differences by sex in the analysis before/after the results obtained on either State STAI or Total STAI.

The main variables (Trait STAI, State STAI, and Total STAI) did not differ significantly in relation to the type of degree, the level of prior knowledge of CPR, or the course studied by the subjects.

As observed in Table 2, statistically significant differences appear between the study groups for anxiety score before the simulation, but not after the simulation. In general, the STAI state increased from 14.7 ± 9.63 points before the simulation to 15.3 ± 7.96 points (*p* > 0.05) afterwards. The Total STAI also did not differ significantly, as it decreased from 10.4 ± 9.01 points to 8.5 ± 7.10 points (*p* > 0.05). 

The respiratory rate revealed significant differences (*p* < 0.05): the experimental heat group had 3.1 ± 1.25 breaths per minute more than the control, and 4.2 ± 1.25 breaths per minute more than the cold group. The cold group did not differ significantly from the control group (*p* > 0.05).

The Trait STAI (basal anxiety) revealed a statistically significant difference of 9.7 ± 3.39 points (*p* < 0.05) between the heat group and the control group. The difference between the cold group and the control group was 5.6 ± 3.25 points, but this was not statistically significant (*p* > 0.05).

Additionally, we analyzed the differences between before/after the State STAI and Total STAI by experimental groups. The decrease was significant on control group in State STAI (*p* < 0.05) and Total STAI (*p* < 0.05). Any of the changes registered in the heat group and cold group was significant (*p* > 0.05).

In general, the State STAI (emotional anxiety) increased by 0.6 ± 1.67 points globally (*n* = 59). When differentiating between groups, we perceived the effect of heat in the State STAI, which was 4.1 ± 3.03 points higher than the control group (*p* < 0.05). However, the cold group differed more from the control group, by 9.7 ± 3.00 points (*p* < 0.05). We also noted statistically significant differences (*p* < 0.05) between the heat and cold group. In the Total STAI, heat produced an increase of 14.3 ± 2.45 points (*p* < 0.05) from the control, and the cold group scored 12.4 ± 2.29 points (*p* < 0.05) more than the control.

After the simulation, there were also differences, albeit not statistically significant. In the hot environment, the STAI score decreased by 4.9 ± 5.59 points compared to the control group, while the cold and control groups differed by 0.3 ± 2.52 points.

These decreases in the State STAI produced a decrease in the Total STAI by 1.9 ± 8.73 points (*p* > 0.05), although without statistically significant differences. There were also no statistically significant differences between experimental groups with decreases of 4.8 ± 2.45 points for heat and 3.9 ± 2.32 points for cold versus the control group, respectively. The before and after test scores according to experimental groups differed only for the heat group that showed an increase of 1.1 ± 6.06 points in the Total STAI.

Before doing CPR, physiological parameters at basal level in individuals with anxiety (STAI-before test) were, in general, lower than those from individuals without anxiety (Table 3). There were statistically significant differences in lactate (1.46 ± 0.68 mmol/L in individuals with anxiety versus 2.16 ± 1.24 mmol/L in individuals without anxiety (*p* < 0.05)) and heart rate (74.71 ± 11.30 bpm in participants with anxiety compared to 83.18 ± 15.02 bpm in participants without anxiety (*p* < 0.05)). There were no differences in the rest of values during the CPR.

There were no statistically significant differences in physiological parameters, measured during the resuscitation, between individuals who presented anxiety after the experience (STAI-after) and those who did not present anxiety. A significant (*p* < 0.05) difference was found in lactate measurement during Minute 9: 5.42 ± 4.00 mmol/L in individuals without anxiety and 3.30 ± 1.70 mmol/L in individuals with anxiety (Figure 2). It must be outlined that subjects who ended the CPR without anxiety had better recovery after the event despite showing higher fatigue signs during the test.

Finally, the response about the quality of CPR presented irregular outcomes in the depth of compressions between the group with anxiety after the test, and the group without it. While Figure 3 shows that total number of compressions per minute (Figure 3a) and frequency of compressions (Figure 3b) were, in general, higher in individuals who presented anxiety after the test (STAI-after), there were no significantly differences. If we analyze the matter in depth, the increase of total number of compressions made by subjects with anxiety from Minute 7 caused a reduction in the percentage of compressions made in optimal frequency (100−120 compressions/minute) in this group (Figure 3c), reducing it under 50% until the end of CPR (49.87% in Minute 7; 47.01% in Minute 8; 49.35% in Minute 9; and 46.16% in Minute 10). There were no statistical differences in quality of CPR performed, in relation to the type of degree, the level of prior knowledge of CPR, or the course studied by the subjects.

## 4. Discussion

One of our main findings was the increase in state anxiety globally by 0.6 ± 1.67 points in situations of performing CPR in extreme temperatures. While analyzing by scenarios, we see that the increase in state anxiety is significant in the control group. These results are in line with other studies in which anxiety was measured during clinical simulation maneuvers in students of the health sciences in stable and controlled room temperature. These studies, whether they used the STAI [22] or other questionnaires [23], coincide in the increase in anxiety without finding significant differences between the sexes [23]. In our case, the only statistically significant difference according to sex was the difference in Total STAI after the simulation, with men displaying less anxiety than women.

Attitudes and beliefs of nursing students regarding CPR seem to be comparable in the first and last years, according to studies that included them as variables [24]. Their results are consistent with the anxiety levels obtained in this study, without significant differences in the main variables according to the type of degree, the level of knowledge of CPR, or the course studied. Other authors, however, have demonstrated that an inability to intervene in CPR was significantly related to anxiety during resuscitation, and that nursing students clearly expressed a greater need for practices in CPR during their undergraduate period, compared to students of other degrees such as medicine [24].

In addition, our study provides previously unpublished data that the effect of heat influences the State STAI. We observed an increase of 4.1 ± 3.03 points compared to the control group (*p*<0.05), meaning that a higher level of emotional anxiety was perceived among students who performed the maneuvers at temperatures above 40 °C. In addition, their respiratory rate increased considerably. Previous studies observed an increase of physiological variables accompanying anxiety when performing CPR maneuvers [25,26,27], which coincides with our results, to which we add that the effect of heat is greater than that of cold at the physiological level during the CPR. This differential effect would have to be considered when preparing people for interventions in warm temperatures.

Furthermore, individuals who showed anxiety after the CPR (STAI after) presented less lactate in measures made during Minutes 6 and 9 of the event. Our results concur with those found by Hermann et al. [28], who compared physical and psychosocial effects in lactate and determined a greater increase of lactate in blood of those who were subjected to a physical stress. According to these authors, in cases in which psychosocial stressors take part, lactate metabolizing could be related with bigger needs of the brain, in order to carry out cognitive activity. 

At the same moment, we registered an increase of compressions per minute, suggesting that the rate increase in resuscitation maneuvers since Minute 6 could be related with the anxiety of the participant. It unleashes worse quality compressions at the final part of the event. Although there is no previous evidence in CPR at this point, this situation could be compared to poorer performance in professional athletes who present a higher level of anxiety [29,30,31]. In addition, we could compare it to the lack of concentration in students before an exam [32] or the capacity for decision making by emergency teams [5,13].

Stress occurs as a result of external pressure, while anxiety occurs as a result of internal pressure. Thus, in this study, we wanted to measure and compare state anxiety and trait anxiety under the influence of a simulation practice with an added stressor. Other studies have been carried out by adding emotional stressors (actors that simulated family stress) during CPR maneuvers [33], but thus far no studies have added thermal stressors, when it should also be considered that cases of cardiopulmonary arrest could occur under adverse cold or hot temperatures. Our results show that the students presented greater trait (baseline) anxiety in the hot environment, with a statistically significant difference of 9.7 ± 3.39 points (*p* < 0.05) between the heat group and the control group.

However, the effects of anxiety on simulation learning are somewhat controversial in the literature. Some authors state that anxiety can lead to an excellent intervention [23], others talk about the close relationship with memory and learning [8], but in more recent reviews the relationship is not clear [8]. The anxiety that occurs in such emergency situations and the differences in some environments and others are evident; some authors [11] have seen how anxiety significantly decreases in subsequent clinical simulations, as well as the capacity for clinical decision-making of students in the face of vital emergency scenarios, but the positive or negative effect of this emotional anxiety on learning at the clinical level remains to be determined in future research. In contrast, some authors reveal that emergency assistance personnel do not use structured psychological coping strategies for facing stress situations such as death after CPR [34]. Therefore, it would be important that when applying CPR learning techniques, coping techniques should be included to strengthen the resilience of future professionals. Hypnotherapy has a positive effect on the academic performance and the reduction of anxiety of undergraduate students [35,36,37]. For example, giving a testimony either in writing [38,39,40,41] or through group discussions could help reduce anxiety. These coping therapies could be applied to the teaching of clinical practice.

This study has some limitations in the small sample size and the homogeneity of students in age and educational level, so the results are hardly generalizable. As described on the study design, there was an aleatory distribution of the participants according to sex in the three groups, although many confounding variables could affect the outcomes; for that reason, possible differences found by the type of degree, the level of prior knowledge of CPR, of the course studied by the subject were analyzed. Finally, neither the composition of the groups nor the results showed statistical differences. There may also have been confounding factors in the presence of teachers, limited time, or the simulator’s visualization of the quality of compressions during the test, which may have modified the results somewhat. Besides, as the first study that links CPR, extreme temperature, and anxiety, there is no literature to compare our results.

As strengths, this study has included scenarios of extreme temperatures of cold and heat that involved a complex use of resources and organization, and that reveal results that have not been published previously.

Finally, other clinical simulation studies have used different questionnaires to measure anxiety, such as Nursing Anxiety and Self-confidence with Clinical Decision-Making (NASC-CDM) [11], although the most frequently used in these cases is the STAI, as a validated and effective tool for measuring self-perceived anxiety. In addition, we measured physiological variables that strengthen the results regarding the effects of anxiety.

## 5. Conclusions

Based on this study, we can say that, when faced with extreme adverse temperature scenarios (extreme heat and cold), students with basic knowledge in CPR generated a greater situational anxiety than in situations at room temperature, especially in conditions of extreme heat. Regarding the physiological effects, we found less lactate, before and during the event, in individuals with anxiety, which could determine less quality in compressions of CPR beginning at Minute 7. Hence, programs targeted at promoting coping mechanisms to reduce anxiety before a critical clinic situation should be implemented in health academic training, before these situations arise in their working life.

## Figures and Tables

**Figure 1 ijerph-17-04241-f001:**
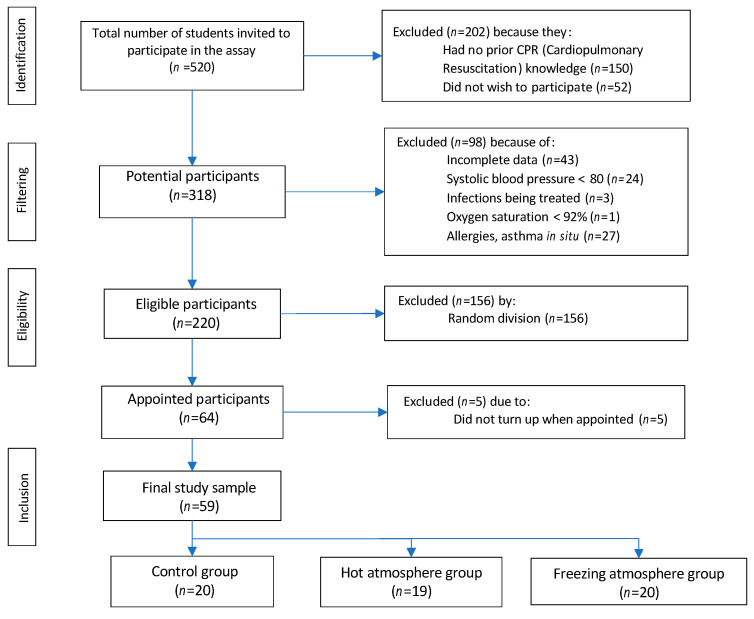
Flow chart of the selection of participants for this study.

**Figure 2 ijerph-17-04241-f002:**
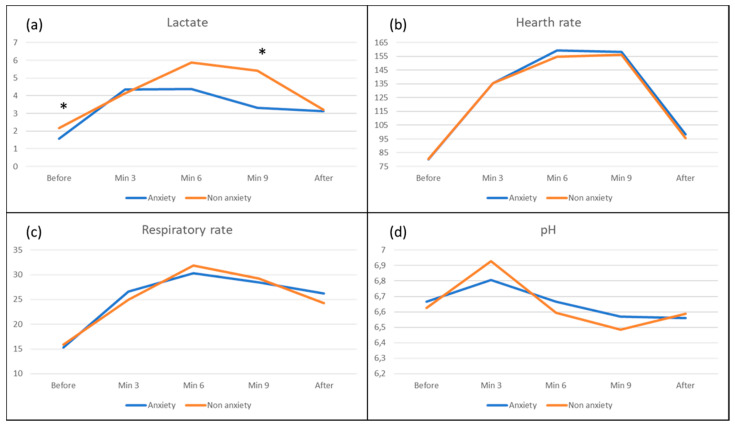
Physiological response during CPR according to anxiety level: (**a**) lactate; (**b**) heart rate; (**c**) respiratory rate; and (**d**) pH. * Statistically significant differences. Lactate Before (*p* = 0.036), Lactate at Minute 9 (*p* = 0.010).

**Figure 3 ijerph-17-04241-f003:**
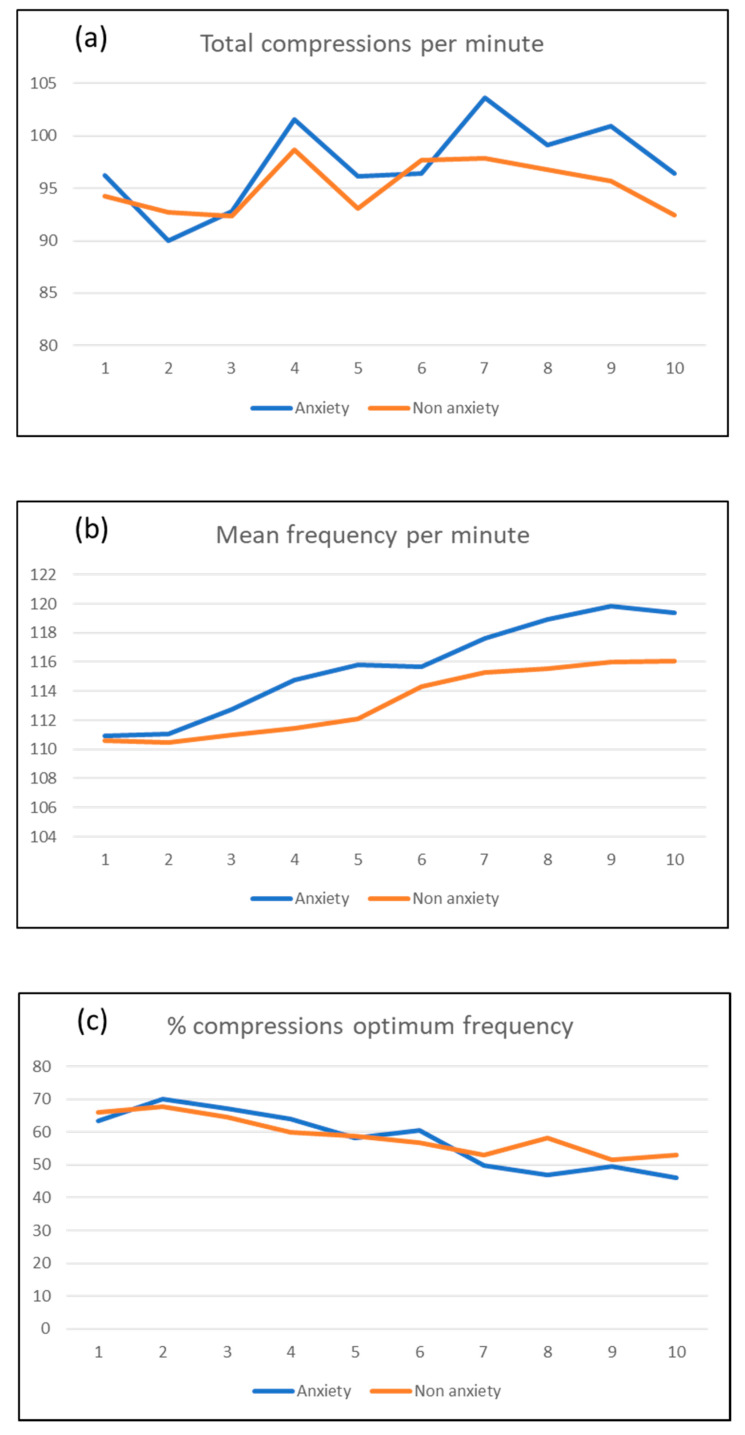
Quality of CPR according to anxiety: (**a**) total compressions per minute; (**b**) mean frequency per minute; and (**c**) present of compressions at optimum frequency.

**Table 1 ijerph-17-04241-t001:** State-Trait Anxiety Inventory for adults, variables according to sex.

Variables	Male		Female		*p* Value Sex
	*n*	Mean ± SD	*n*	Mean ± SD	
Age	27	21.5 ± 2.76	32	21.2 ± 2.83	0.648
Trait STAI	25	18.7 ± 9.69	27	22.3 ± 10.52	0.214
State STAI before	27	12.6 ± 9.16	28	16.7 ± 9.81	0.120
State STAI after	26	16.9 ± 8.20	30	14.0 ± 7.62	0.173
*p* value State STAI before/after		0.068		0.245	
Total STAI before	25	8.7 ± 8.27	26	12.0 ± 9.54	0.187
Total STAI after	25	6.4 ± 5.55	27	10.4 ± 7.89	0.039 *
*p* value Total STAI before/after		0.181		0.401	

* Statistically significant differences. SD, Standard Deviation.

**Table 2 ijerph-17-04241-t002:** Variables according to experimental group.

STAI Variables	Control			Experimental Group				Total		*p* Value
			Heat Group			Cold Group				
	*n*	Mean ± SD	*n*	Mean ± SD		*n*	Mean ± SD	*n*	Mean ± SD	
Trait STAI	16	25.7 ± 11.19	16	16.0 ± 7.60		20	20.1 ± 9.68	52	20.6 ± 10.19	0.022 *
State STAI before	16	9.7 ± 8.94	19	13.9 ± 8.01		20	19.4 ± 9.74	55	14.7 ± 9.63	0.008 *
State STAI after	18	17.0 ± 8.65	18	12.1 ± 8.33		20	16.7 ± 6.28	56	15.3 ± 7.96	0.108
*p* value State STAI before/after		0.005 *		0.364			0.285		0.666	
Total STAI before	15	19.8 ± 9.23	16	5.4 ± 5.16		20	7.4 ± 5.49	51	10.4 ± 9.01	0.000 *
Total STAI after	16	11.5 ± 6.89	16	6.6 ± 7.02		20	7.6 ± 6.88	52	8.5 ± 7.10	0.116
*p* value Total STAI before/after		0.008 *		0.446			0.905		0.122	

* Statistically significant differences. SD, Standard Deviation. STAI, State-Trait Anxiety Inventory.

**Table 3 ijerph-17-04241-t003:** Physiological variables before test according to anxiety level (STAI before).

STAI Before	Anxiety		Non-Anxiety		*p* Value
	*n*	Mean ± SD	*n*	Mean ± SD	
Lactate before	17	1.46 ± 0.68	34	2.16 ± 1.24	0.012 *
Hearth rate before	17	74.71 ± 11.30	34	83.18 ± 15.02	0.030 *
Respiratory rate before	17	14.59 ± 2.74	34	16.32 ± 5.06	0.119
Temperature before	17	36.59 ± 0.48	34	36.75 ± 0.35	0.181
pH before	17	6.72 ± 0.40	34	6.66 ± 0.58	0.716

* Statistically significant differences. SD, Standard Deviation.

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
