# Peer review of "Do Rescuers’ Physiological Responses and Anxiety Influence Quality Resuscitation under Extreme Temperatures?"

_ijerph, 2020, doi:10.3390/ijerph17124241_

Round 1

Reviewer 1 Report

The study seems very interesting and the topic addressed considers important aspects that have to be considered during the education of health professionals.

I have some comments that I hope it will help to clarify some aspects and improve the manuscript.

Abstract: Please include  the interest in evaluating the effect of the extreme temperatures on the objective of the study (line 20-21)

Introduction: Briefly justify why the extreme temperatures are relevant. What is the hypothesis proposed?

Study design: Give some reference to the descriptive cross-sectional before-after design. The methodology seemed to better accommodate a randomized clinical trial.

It is not clear what were the criteria to consider an eligible person in relation to age (18-30 or 18-65)

Could you better specify the exclusion criteria related to temperature? (line 84 - was there a cutoff point?

In the sample size, clarify the number of each group (18 participants x 3=  54) (line 86)

Line 94: in relation to the phrase: "assigned with double blind masking". Please clarify what elements were double blind masking and how this process was guaranteed.

Results

Line 153: in relation to the phrase: "Most (72.9%, n: 43) of the subjects already had basic knowledge of CPR". Did it seem that the basic knowledge of CPR was an inclusion criteria? or it was evaluated in a different way?

Table 2- line 169: it seems to me that there is a writing error: “The total STAI 168 also did not differ significantly as it decreased from 10.4 ± 9.01 points to 5.5 ± 7.10 points (p> 0.05)” It seems to me that the figure is 8.5 no 5.5

Discusion

I find it interesting to include in the discusion the variation that each group had before and after the intervention (each being its own control), specially because the limitation of controlling for the base variables that each group has. In this sense, I find it interesting to discuss the variation before / after / by sex, and thus also in Table 2, where for example in the extreme temperature group there was less variation than in the control group. In this group the Total STAI variation decreased by 8.3 while the extreme heat group increased 1.2 and the cold group increased by 0.2.

Consider in the discussion the effect of other variables that could affect the interpretation of the results as other confounding variables not considered. This in particular because there are already some differences in the baseline of the groups (foe example having taken a previous simulation exam). Thus, it would also be interesting to discuss whether the distribution by sex in the three groups and their characteristics are similar in the 3 groups.

Also will be very good to include in the discussion the limitation of the sample size and the statistical tests applied. Was the use of non-parametric statistical tests not required?

Author Response

Reply to the comments on manuscript “Do rescuers' physiological responses and anxiety influence quality resuscitation under extreme temperatures?”

[ijerph-821457].

We wish to thank you all for your constructive comments in this round of review. Your comments provided valuable insights to refine its contents and analysis. In this document, we try to address the issues raised as best as possible.

Reply to Reviewer 1:

  • Abstract: Please include the interest in evaluating the effect of the extreme temperatures on the objective of the study (line 20-21)

  • According to reviewers´ comments, the following changes were included in the abstract:

“The aim of this study was to assess the physiological effects and anxiety levels of health sciences undergraduates when faced with CPR process in different temperatures (room temperature, extremely cold or extremely warm).” (line 20-22)

  • Introduction: Briefly justify why the extreme temperatures are relevant. What is the hypothesis proposed?

  • According to reviewers´ suggestion, the following changes were included in the Introduction:

“So far, no studies have been conducted in adverse temperature conditions of cold and heat. It is unknown how adverse temperature could affect the CPR process, and the derived anxiety on rescuers. This could be relevant for people who perform CPR in geographic places with adverse climate. We based our study on the possible differences between physiological effects and anxiety levels of rescuers performing CPR in extremely temperature vs room temperature.” (line 61-66)

  • Study design: Give some reference to the descriptive cross-sectional before-after design. The methodology seemed to better accommodate a randomized clinical trial.

  • In relation to the study design, we need to concrete that this is not a randomized clinical trial: there were no intervention on the group. We just registered data in different cohorts.

  • It is not clear what were the criteria to consider an eligible person in relation to age (18-30 or 18-65)
    • Regarding to the inclusion criteria, the age of possible adult participants could be in between 18-65, although effectively, the final selected group were among 18 and 30 years old.

Thus, according to the reviewer´ comment, we corrected the error and we included the following explanation:

“All selected participants aged between 18 and 30. They all had basic knowledge of CPR process, according to the American Heart Association or European Resucitation Council training.” (line 76-78)

  • Could you better specify the exclusion criteria related to temperature? (line 84 - was there a cutoff point?

  • Thanking the reviewer, we modified the sentence in line 93. As an exclusion criteria we considered the participant temperature when it was beyond 38ºC.

  • In the sample size, clarify the number of each group (18 participants x 3= 54) (line 86)

  • Effectively, it was an error on line 86 (now line 95), that has been corrected on the final manuscript:

“The sample size were 18 subjects per group (54 subjects)”

  • Line 94: in relation to the phrase: "assigned with double blind masking". Please clarify what elements were double blind masking and how this process was guaranteed.

  • In this point, we would like to clarify that participants didn´t know in what group they were allocated for the study, not to disrupt psychological neither physiological reaction on the subjects. Besides, the researcher who made the analysis didn´t know to which group belongs each individual; we worked with encode variables.

Thanking the reviewer, this issue is now addressed in the revised version of the paper:

“Three homogeneous groups were assigned with double blind masking: none participant knew in what kind of environment was going to perform the test. The variable of temperature was encoded in order to guarantee the blind masking process.” (line 102-105)

About Results:

  • Line 153: in relation to the phrase: "Most (72.9%, n: 43) of the subjects already had basic knowledge of CPR". Did it seem that the basic knowledge of CPR was an inclusion criteria? or it was evaluated in a different way?

  • Maybe the language is not clear enough at this point. The fact is that 72.9% of participants had previous knowledge about CPR, and the remaining 27.1% had to take a specific training at University, in order to keep the requirements (inclusion criteria). Hence, this sentence was included on the reviewed paper:

“The 72.9% of participants had previous knowledge about CPR, and the remaining 27.1% had to take a specific training at University, in order to keep the requirements or inclusion criteria. There were no statistical differences in the quality of CPR performed by both groups.” (line 164-166)

  • Table 2- line 169: it seems to me that there is a writing error: “The total STAI 168 also did not differ significantly as it decreased from 10.4 ± 9.01 points to 5.5 ± 7.10 points (p> 0.05)” It seems to me that the figure is 8.5 no 5.5

  • Effectively, there´s an error on line 169. We thank the meticulousness of reviewers, therefore it has been corrected on the new paper:

“The total STAI also did not differ significantly as it decreased from 10.4 ± 9.01 points to 8.5 ± 7.10 points (p> 0.05).” (line 182)

About Discussion:

  • I find it interesting to include in the discussion the variation that each group had before and after the intervention (each being its own control), specially because the limitation of controlling for the base variables that each group has. In this sense, I find it interesting to discuss the variation before / after / by sex, and thus also in Table 2, where for example in the extreme temperature group there was less variation than in the control group. In this group the Total STAI variation decreased by 8.3 while the extreme heat group increased 1.2 and the cold group increased by 0.2.

  • According to the reviewer comments, we have considered to include the differences before/after by sex in Table 1 (line 178), and by experimental groups in Table 2 (line 194) as well. Thus, the changes in the revised manuscript are as follows:

“There were no statistical differences by sex in the analysis before/after the results obtained on State STAI either on Total STAI.” (line 174-175)

“Additionally, we analyzed the differences between before/after the State STAI and Total STAI by experimental groups. The decrease was significant on control group in State STAI (p<0.05) and Total STAI (p<0.05). Any of the changes registered in the Heat group and Cold group was significant (p>0.05).” (line 190-193)

“One of our main findings was the increase in state anxiety globally by 0.6 ± 1.67 points in situations of performing CPR in extreme temperatures. While analysing by scenarios, we see that the increase on state anxiety is significant on the control group. These results are in line with other studies in which anxiety was measured during clinical simulation maneuvers in students of the health sciences in stable and controlled room temperature.” (line 244-248)

  • Consider in the discussion the effect of other variables that could affect the interpretation of the results as other confounding variables not considered. This in particular because there are already some differences in the baseline of the groups (for example having taken a previous simulation exam). Thus, it would also be interesting to discuss whether the distribution by sex in the three groups and their characteristics are similar in the 3 groups.

  • Many variables that could be confounding variables were previously analyzed before sending the manuscript, but there were no statistical differences so finally, there were no included on final version. Indeed, we didn’t find statistical differences in relation to the type of degree, the level of prior knowledge of CPR, as well as the course studied by the subjects. According to the reviewer suggestion, we added an explanation on line 303-309.

This study has some limitations in the small sample size and the homogeneity of students in age and educational level, so the results are hardly generalizable. As described on the study design, there was an aleatory distribution of the participants according to sex in the three groups, although many confounding variables could affect to the outcomes; for that reason, there were analyzed possible differences found by the type of degree, the level of prior knowledge of CPR, as well as the course studied by the subject. Finally, neither the composition of the groups nor the results shown statistical differences. “

  • Also will be very good to include in the discussion the limitation of the sample size and the statistical tests applied. Was the use of non-parametric statistical tests not required?

As mentioned on the discussion, it was a small and homogeneous sample on this study, which makes that the outcomes can´t be generalised (line 303-304).

With respect to the use of non parametric tests, we mentioned in section of “data analysis” that “Data were checked to meet the normality condition by the Kolmogorov-Smirnov test”; there was not necessary to use non- parametrics tests due to all the data presented a normal distribution.

Taking the point of the reviewer, we added on the paper the following clarification:

“All the data presented a normal distribution, hence non parametric tests were not necessary.” (line 152-153)

Once again, we thank you for the time you put in reviewing our paper and look forward to meeting your expectations. Since your inputs have been precious, in the eventuality of a publication, we would like to acknowledge your contribution explicitly.

The authors’

Reviewer 2 Report

I appreciate the thoughtful work of the researchers in elucidating the nuances of CPR training and performance optimization process. 

Line 19-20: May modify the sentence as “although CPR is taught as a simulation, it can still be stressful for the trainees since it mimics/resembles a real-life circumstances” 

Line 20-21: “... physiological effects and anxiety levels that CPR” Did you mean “of the CPR process”?

Line 27-28: What do the scores signify? May a short detail be added?

Line 28-30: The sentence “Furthermore,... since minute 7” needs modification, for example, “... less quantity of CPR in minute 7”. Does this refer to CPR performance duration of trainees, or does this mean at 7 minutes a significant decrease in performance of trainees was observed?

Line 39: “... than in the classroom…”. Please clearly define based on what parameters you are distinguishing CPR training, that is itself simulated, versus the training outputs in classroom/laboratories. 

Line 76: “pull of”. Maybe it’s “pool of”?

Line 161: What are the “initial phase” and “final and subsequent phase”? Does this refer determining anxiety via CPR, or the STAI questionnaire, or the physiological tests?

Line 162: Define “main variables”.

Line 164: In Table 1, can the mean difference of STAI scores at before and after be displayed, rather than separately?

Line 178: Define “globally”. Is it population in general or the sample?

Line 211-219: It would be better if the performance is reported against the standard number of compression per minute, etc.

Further I have the following questions which might strengthen the paper. Was the level of previous actual/simulation CPR experience show change in CPR performance?

Author Response

Reply to the comments on manuscript “Do rescuers' physiological responses and anxiety influence quality resuscitation under extreme temperatures?”

[ijerph-821457].

Response to Reviewer 2:

We appreciate the thoughtful work of the researchers in elucidating the nuances of CPR training and performance optimization process. Responding to each point:

  • Line 19-20: May modify the sentence as “although CPR is taught as a simulation, it can still be stressful for the trainees since it mimics/resembles a real-life circumstances”

  • We accept the advice and we modify the sentence in line 19-20.

  • Line 20-21: “... physiological effects and anxiety levels that CPR” Did you mean “of the CPR process”?

  • Effectively, we mean exactly that; so we corrected the sentence as follows:

“The aim of this study was to assess the physiological effects and anxiety levels of health sciences undergraduates when faced with CPR process in different temperatures (room temperature, extremely cold or extremely warm).” (line 20-22)

  • Line 27-28: What do the scores signify? May a short detail be added?

  • We refer to the “STAI after CPR score”, that appear defined on the study protocol, in material and methods section. We accept the suggestion of the reviewer, so we have added that term on line 29-30.

  • Line 28-30: The sentence “Furthermore,... since minute 7” needs modification, for example, “... less quantity of CPR in minute 7”. Does this refer to CPR performance duration of trainees, or does this mean at 7 minutes a significant decrease in performance of trainees was observed?

  • We refer to “since minute 7 we observed a remarkable decrease (but not significant) on the performance of rescuers with anxiety”. Accepting the reviewer comments, we added that concrete sentence on line 30-32:

“Furthermore, there was less lactate in blood, before and during the event in individuals with anxiety. In addition, since minute 7 we observed a remarkable decrease (but not significant) on the performance of rescuers with anxiety.”

  • Line 39: “... than in the classroom…”. Please clearly define based on what parameters you are distinguishing CPR training, that is itself simulated, versus the training outputs in classroom/laboratories.

  • In order to improve and clarify that point, we added the following explanation:

“Clinical simulation laboratories provide real scenarios in contrast to conventional lecture rooms or other laboratories, because the available material is more advanced and the feedback with students is immediate [2]. What is more, a CPR simulation can be a stressful experience because it resembles a vital situation of extreme urgency, causing a considerable great anxiety for undergraduates, further than a basic training of theoretical concepts going with practical maneuvers (simulating learning) [3-5].” (line 40-45)

  • Line 76: “pull of”. Maybe it’s “pool of”?

  • It has been corrected on the reviewed paper. (line 85)

  • Line 161: What are the “initial phase” and “final and subsequent phase”? Does this refer determining anxiety via CPR, or the STAI questionnaire, or the physiological tests?

  • We refer to the anxiety according to the STAI questionnaire results. In concrete, “initial phase” refers to the results of “Total STAI before CPR” and “final and subsequent phase” means the results of “Total STAI after CPR”. We have modified that on line 172-174.

  • Line 162: Define “main variables”.

  • Accepting the advice, on reviewed manuscript we clarify that: “The main variables (Trait STAI, State STAI, Total STAI)”. (line 176)

  • Line 164: In Table 1, can the mean difference of STAI scores at before and after be displayed, rather than separately?

  • According to reviewers´ comments, we have modified the position of results in Table 1 in order to show the results before and after consecutively in each category. Besides, we added the p value of the before/after difference in table 1 (line 178) and 2 (line 194).

  • Line 178: Define “globally”. Is it population in general or the sample?

  • Effectively, we referred to the sample studied (control group + heat group + cold group). So we outlined a clarification “n=59” on line 196.

  • Line 211-219: It would be better if the performance is reported against the standard number of compression per minute, etc.

  • We accept the reviewer comment so we have outlined the number of standard compressions per minute (100-120) on line 235.

  • Further I have the following questions which might strengthen the paper. Was the level of previous actual/simulation CPR experience show change in CPR performance?

  • Actually, it has been analyzed but we didn´t find significant differences, hence we didn´t included in the first manuscript. However, now we have added an explanation, indicating that:

There were no statistical differences in quality of CPR performed, in relation to the type of degree, the level of prior knowledge of CPR, as well as the course studied by the subjects” (line 237-239)

Once again, we thank you for the time you put in reviewing our paper and look forward to meeting your expectations. Since your inputs have been precious, in the eventuality of a publication, we would like to acknowledge your contribution explicitly.

The authors’

Round 2

Reviewer 1 Report

Thank you very much for all the work and for taking into account the observations and suggestions in the second version of the article!

You have addressed and clarified some observed aspects so that readers can understand it without difficulty.

A minor comment would be to include a bibliographic reference to support the study design mentioned, since there could be some differences with other people who read your article. It could be interpreted as a randomized study considering that the application of PCR was the intervention randomly assigned to the participants.

Author Response

Reply to the comments on manuscript “Do rescuers' physiological responses and anxiety influence quality resuscitation under extreme temperatures?”

[ijerph-821457].

We wish to thank you all for your constructive comments in this round of review. Your comments provided valuable insights to refine its contents and analysis. In this document, we try to address the issues raised as best as possible.

Reply to Reviewer 1( Round 2)

A minor comment would be to include a bibliographic reference to support the study design mentioned, since there could be some differences with other people who read your article. It could be interpreted as a randomized study considering that the application of PCR was the intervention randomly assigned to the participants.

  • According to reviewer suggestion, the following bibliographic reference were included to support study desing
